# Protein Levels of Anti-Apoptotic Mcl-1 and the Deubiquitinase USP9x Are Cooperatively Upregulated during Prostate Cancer Progression and Limit Response of Prostate Cancer Cells to Radiotherapy

**DOI:** 10.3390/cancers15092496

**Published:** 2023-04-26

**Authors:** Sophia A. Hogh-Binder, Diana Klein, Frederik Wolfsperger, Stephan M. Huber, Jörg Hennenlotter, Arnulf Stenzl, Justine Rudner

**Affiliations:** 1Department of Radiation Oncology, University Hospital Tuebingen, Hoppe-Seyler-Str. 3, 72076 Tuebingen, Germany; 2Institute for Cell Biology (Cancer Research), University Hospital Essen, University of Duisburg-Essen, Virchowstr. 173, 45147 Essen, Germany; 3Department of Urology, University Hospital Tuebingen, Hoppe-Seyler-Str. 3, 72076 Tuebingen, Germany

**Keywords:** prostate cancer, radiotherapy, Mcl-1, protein stability, ubiquitylation, USP9x

## Abstract

**Simple Summary:**

Radiotherapy constitutes an important therapeutic option for prostate cancer. However, prostate cancer cells often acquire resistance during cancer progression, limiting the cytotoxic effects of radiotherapy. Among factors regulating sensitivity to radiotherapy are members of the Bcl-2 protein family, known to regulate apoptosis at the mitochondrial level. We demonstrate that protein levels of deubiquitinase USP9x and anti-apoptotic Mcl-1 increased during prostate cancer progression. Downregulation of Mcl-1 or USP9x levels improved the response of prostate cancer cells to radiotherapy. Moreover, radiotherapy itself was able to regulate Mcl-1 protein stability in prostate cancer cells.

**Abstract:**

Background: Radiotherapy constitutes an important therapeutic option for prostate cancer. However, prostate cancer cells often acquire resistance during cancer progression, limiting the cytotoxic effects of radiotherapy. Among factors regulating sensitivity to radiotherapy are members of the Bcl-2 protein family, known to regulate apoptosis at the mitochondrial level. Here, we analyzed the role of anti-apoptotic Mcl-1 and USP9x, a deubiquitinase stabilizing Mcl-1 protein levels, in prostate cancer progression and response to radiotherapy. Methods: Changes in Mcl-1 and USP9x levels during prostate cancer progression were determined by immunohistochemistry. Neutralization of Mcl-1 and USP9x was achieved by siRNA-mediated knockdown. We analyzed Mcl-1 stability after translational inhibition by cycloheximide. Cell death was determined by flow cytometry using an exclusion assay of mitochondrial membrane potential-sensitive dye. Changes in the clonogenic potential were examined by colony formation assay. Results: Protein levels of Mcl-1 and USP9x increased during prostate cancer progression, and high protein levels correlated with advanced prostate cancer stages. The stability of Mcl-1 reflected Mcl-1 protein levels in LNCaP and PC3 prostate cancer cells. Moreover, radiotherapy itself affected Mcl-1 protein turnover in prostate cancer cells. Particularly in LNCaP cells, the knockdown of USP9x expression reduced Mcl-1 protein levels and increased sensitivity to radiotherapy. Conclusion: Posttranslational regulation of protein stability was often responsible for high protein levels of Mcl-1. Moreover, we demonstrated that deubiquitinase USP9x as a factor regulating Mcl-1 levels in prostate cancer cells, thus limiting cytotoxic response to radiotherapy.

## 1. Introduction

Prostate cancer is one of the most prevalent malignancies and the fifth most common cancer death cause in men [1]. Therapeutical interventions include surgical resection, chemotherapy, hormone ablation therapy, radiotherapy, or just surveillance. The latter usually applies to tumors in an early stage with very slow progression or when the life expectancy of the patient is low [2,3,4,5]. In contrast, radiotherapy is usually applied to prostate cancer in advanced stages. Despite the efforts and advances in prostate cancer treatment, the treatment of patients with advanced prostate cancer remains challenging. A reason for the limited therapeutical response, particularly in advanced prostate cancer, is an increased cell death resistance to chemotherapeutic drugs and radiotherapy [4,6].

Mitochondria play a central role in therapy resistance, linking signals essential for cell survival and proliferation, such as energy production, to cell death. B cell leukemia-2 (Bcl-2) family members control mitochondrial morphology, mitochondrial quality (by mitophagy), mitochondrial calcium levels, and mitochondrial outer membrane permeabilization (MOMP) for apoptotic factors in the intrinsic apoptosis pathway [7,8,9]. The Bcl-2 protein family is divided into anti-apoptotic and pro-apoptotic groups according to structural and functional similarities. While anti-apoptotic Bcl-2 family members prevent MOMP, pro-apoptotic Bcl-2 proteins facilitate MOMP and, thereby, cytochrome C release from mitochondria into the cytosol through pores in the outer mitochondrial membrane, where cytochrome C promotes the assembly of the apoptosome, a heptameric pro-apoptotic complex [10]. At the complex, the initiator caspase-9 becomes activated and, in turn, activates executor caspases to be proteolytic cleavage, upon which the cells dismantle, giving rise to typical apoptotic morphology. The change in Bcl-2 rheostat, the balance between pro- and anti-apoptotic Bcl-2 family members, regulates the sensitivity to apoptosis.

Levels of anti-apoptotic proteins Bcl-2, Bcl-xL, and Mcl-1 are upregulated in prostate cancer and contribute to prostate cancer progression [11,12,13]. All three proteins limit the cytotoxic response of prostate cancer cells to anti-cancer therapies [14,15,16,17].

In contrast to the anti-apoptotic proteins Bcl-2 and Bcl-xL, the anti-apoptotic protein Mcl-1 is characterized by a rapid turnover in normal tissue [18,19]. Protein levels of the anti-apoptotic Mcl-1 are regulated on a transcriptional, translational, and posttranslational level [20,21,22]. In prostate cancer cells, the transcription factors β-catenin and hypoxia-inducible factor (HIF)-1α were able to upregulate MCL1 gene expression [23,24,25], while specific miRNAs were able to downregulate MCL1 mRNA and thus also Mcl-1 protein levels [26,27]. Mcl-1 usually displays a short half-life time in non-malignant cells. The protein stability of Mcl-1 is controlled by ubiquitin ligases, which attach polyubiquitin chains to the protein, thus marking it for proteasomal degradation. In this context, β-transducin repeats-containing protein (β-TRCP) and F-box and WD repeat domain-containing 7 (FBW7), both subunits of the Skp/Cullin/F-box (SCF) E3 ubiquitin ligase, as well as the homologous to E6-AP carboxyl terminus (HECT) domain-containing E3 ligase HUWE-1 (also known as Mcl-1 ubiquitin *ligase* E3 (MULE)) are able to interact with Mcl-1 and facilitate its ubiquitylation [28,29,30,31]. On the other hand, the deubiquitinase USP9x removes polyubiquitin chains from Mcl-1, thus preventing degradation of the anti-apoptotic protein and increasing its half-life time [22,29,32,33]. Moreover, phosphorylation of Mcl-1 at serine 159 and threonine 163 promotes its ubiquitylation and reduces protein stability [34].

Here, we evaluated protein levels of Mcl-1 and USP9x, a deubiquitinase regulating Mcl-1 protein stability, in human tissue samples during prostate cancer progression. In addition, we analyzed how levels of the protective protein Mcl-1 were regulated in prostate cancer cells. In particular, we examined the posttranslational regulation of Mcl-1 protein levels by USP9x and the relevance of this mechanism for cell survival in human LNCaP and PC3 prostate cancer cells. We further analyzed whether and how radiotherapy, which is commonly applied to patients with progressed prostate cancer, affected Mcl-1 protein levels and its turnover in LNCaP and PC3 cells.

## 2. Materials and Methods

### 2.1. Reagents and Antibodies

Cycloheximide was purchased from Sigma (Deisenhofen, Germany). ABT-263 was obtained from Active Biochemicals (Bonn, Germany). The following antibodies were used for Western blotting: rabbit-anti Bak from Upstate and mouse-anti Noxa from Calbiochem (Millipore, Schwalbach, Germany), rabbit-anti caspase-3, PARP, Mcl-1, phospho-Mcl-1 (Ser159/Thr163), Bcl-xL, Bax, Bak and Tubulin from Cell Signaling (NEB, Frankfurt, Germany), rabbit-anti Puma and rabbit-anti Bim from Epitomics (Biomol, Hamburg, Germany),mouse-anti Bcl-2 from Santa Cruz Biotechnology (Heidelberg, Germany), rabbit-anti USP9x from Novus Biologicals (Acris, Heford, Germany), mouse anti-Noxa from Calbiochem (Merck, Darmstadt, Germany), mouse-anti GAPDH from Abcam (Cambridge, UK), and mouse-anti β-actin was obtained from Sigma (Deisenhofen, Germany).

### 2.2. Cells and Cell Culture

LNCaP and PC3 prostate cancer cell lines were from ATCC (Bethesda, MA, USA). Cells were grown in RPMI 1640 medium supplemented with 10% fetal bovine serum (Gibco Life Technologies, Eggenstein, Germany) and maintained in a humidified incubator at 37 °C and 5% CO_2_. Cells were routinely tested for mycoplasma contamination once a month and authenticated by STR profiling. Cells were irradiated at room temperature with 6 MV photons with a linear accelerator (LINAC SL25 Philips, DA Best, Netherlands) at a dose rate of 4 Gy/min.

### 2.3. Transfection with siRNA

3–4 × 10^5^ cells were seeded in 2 mL in a 6-well plate. 24 h later, cells were transfected with the respective siRNA at final concentrations of 10–100 pmol using Trans-IT siQuest^®^ transfection reagent (Mirus, Madison, WI, USA) according to the manufacturer’s protocol. Mcl-1 and USP9x ON-TARGET SMARTpools and siCONTROL NON-TARGETING pool were purchased from Dharmacon (Chicago, IL, USA).

### 2.4. Flow Cytometric Analysis

The mitochondrial membrane potential (ΔΨm) was analyzed using ΔΨm-specific dye tetramethylrhodamine ethyl ester (TMRE, Molecular Probes, Mobitech, Goettingen, Germany). At the indicated time points, cells were stained for 30 min in PBS containing 25 nM TMRE. To examine DNA fragmentation, cells were incubated with PBS containing 0.1% sodium citrate, 0.1% Triton X-100, and 10 µg/mL propidium iodide. Cells were detected in channel 2 employing a FACS Calibur flow cytometer (Becton Dickinson, Heidelberg, Germany) and analyzed with the FCS Express 3 software (De Novo Software, Los Angeles, CA, USA). Data show mean values ± S.D. of at least 6 independent experiments.

### 2.5. Colony Formation Assay

Clonogenic survival was analyzed as described before [33]. In brief, cells were seeded at different dilutions in 6-well plates and transfected with siRNA the next day. 24 h after transfection, cells were irradiated with 0–5 Gy. The cells were incubated for 8–10 days to allow the growth of single colonies. After that, cells were fixed with 3.7% formaldehyde and 70% ethanol and subsequently stained with 0.05% Coomassie Brilliant Blue. Colonies (>50 cells/colony) were counted. To determine the survival fraction (SF), the ratio of colonies counted/seeded cells was calculated and normalized to that of untreated controls. The fitting of the curves was performed using Excel software. Error bars indicate the mean values ± SD of 3 independent experiments.

### 2.6. Western Blot Analysis

Cells were lysed in lysis buffer containing 50 mM HEPES pH 7.5, 150 mM NaCl, 1% Triton X-100, 1 mM EDTA, 10 mM sodium pyrophosphate, 10 mM NaF, 2 mM Na3VO4, 100 mM PMSF, 5 µg/mL Aprotinin, 5 µg/mL Leupeptin, and 3 µg/mL Pepstatin A. Protein was separated by SDS-PAGE under reducing conditions and transferred onto PVDF membranes (Roth, Karlsruhe, Germany). Blots were blocked in TBS buffer containing 0.05% Tween 20 and 5% non-fat dry milk for 1 h at room temperature. The membrane was incubated overnight at 4 °C with the respective primary antibodies. The secondary antibody was incubated for 1 h at room temperature. Detection of antibody binding was performed by enhanced chemiluminescence (ECL Western blotting analysis system, GE Healthcare, Freiburg, Germany). Equal loading was verified by detecting β-actin or tubulin. Where indicated, protein levels were quantified by densitometry using ImageJ software (ImageJ 1.40 g NIH, Madison, WI, USA). All Western blot experiments were repeated at least once in independent experiments.

### 2.7. Analysis of Mcl-1 and USP9x Gene Expression in Human Prostate Samples

Gene expression of MCL1 and USP9x was analyzed in normal prostate tissue samples (N: n = 52) and prostate adenocarcinoma tissue samples (T: n = 492) using TCGA data sets at the GEPIA platform (http://gepia.cancer-pku.cn). For differential analysis, expression data were log_2_ (TPM + 1) transformed. log_2_FC is defined as median gene expression in tumor tissues minus median gene expression in normal tissue (median(T)—median(N)). Statistical significance was calculated using one-way ANOVA. Gene expression with log_2_FC > 1 and *p* < 0.01 are considered differentially expressed. Results were presented in Box plots.

The relationship between the gene of interest (MCL1 or USP9X) and survival was assessed using the TCGA dataset generated by Cancer Genome Atlas Research Network [35] at Prostate Integrative Expression Database (https://pixdb.org.uk/PIXdb/pages/index.php, accessed on 7 March 2023). To calculate significance, a univariate model was applied to the survival data. Samples were assigned to risk groups based on the median dichotomization of mRNA expression intensities of the respective gene. Relationships are presented as Kaplan-Meier plots.

### 2.8. Analysis of Mcl-1 and USP9x Protein Levels in Human Prostate Samples

Human tissue samples were obtained from the department of Urology, University of Tuebingen, with patients’ consent approved by the local ethic committee (No. 517/2012BO2 und 379/2010BO1). Resected tissue specimens were processed for pathological diagnostic routine in agreement with institutional standards, and diagnoses were made based on current WHO criteria. Human tissue samples were analyzed anonymously. IHC was performed as previously described [36,37].

Formalin-fixed, paraffin-embedded human prostate tissue sections (4 µm) were prepared by using a descending alcohol series and incubation with citrate buffer, pH 6.1, as the target retrieval solution. Immunohistochemical staining (overnight incubation at 4 °C) was performed using mouse anti-USP9x from Abnova (Acris, Heford, Germany, 1:800 dilution) and rabbit anti-Mcl-1 (Santa Cruz Biotechnology, Heidelberg, Germany, 1:1600 dilution). Primary antibodies were detected by horseradish–peroxidase-conjugated secondary antibodies and DAB-staining. Nuclei were counterstained with hematoxylin.

Combined quantitative and qualitative evaluation of immunoreactivities were performed blinded to clinical/follow-up data using immunoreactivity scores for absent (0), weak (1), moderate (2) and strong positive (3) protein expression levels. The immunoreactivity score takes into account both the percentage of positive cells and staining intensity [37,38]. The mean ± S.D. and the 95 percentile were calculated for each grade and each antibody staining. To analyze the correlation between Mcl-1 IRS and USP9x IRS, Spearman’s and Pearson’s correlation coefficients (r_s_ and r_p_) were calculated.

### 2.9. Determination of Protein Stability

Cells were treated with 2 µM cycloheximide for 0–3 h. At indicated time points, cells were lysed, and lysates were separated as described above. Mcl-1 protein levels were detected by Western blot. Several blots were made from the same lysates. Protein levels were quantified by densitometry using ImageJ software (ImageJ 1.40 g NIH, Madison, WI, USA) and normalized to the β-actin levels. Then, Mcl-1 levels normalized to the initial level (0 min cycloheximide). Monoexponential decay was fitted using Origin 6.0 software (OriginLab, Northampton, MA, USA). At least three independent experiments were performed.

### 2.10. Data Analysis

Unless otherwise indicated, data were obtained from at least 3 independent experiments. Statistical significance was calculated by Welch-corrected student *t*-test or respective ANOVA test followed by a Bonferroni post-test using GraphPad Software (San Diego, CA, USA).

## 3. Results

### 3.1. Mcl-1 Protein Level but Not Gene Expression of MCL1 Is Elevated in Human Prostate Cancer Samples

Gene expression is commonly used to predict the behavior of tumor cells in response to radiotherapy or chemotherapies. We, therefore, compared gene expression of *MCL1* in normal prostate samples (n = 52) and prostate cancer samples (n = 492) using data sets at the GEPIA platform. *MCL1* gene was expressed at similar levels in normal prostate and prostate cancer tissue (Figure 1A, left box plot). In addition, we analyzed the expression of *USP9X* coding for the deubiquitinase USP9x, *HUWE1* coding for the MCL-1-specific E3 ligase MULE, *BTRC* and *FBXW7* coding for β-TRCP and FBW7, respectively, both are subunits of the SCF E3 ligase complex recognizing Mcl-1 [28,29,30,31]. No significant differences in gene expression were observed between normal and prostate cancer tissue for any of these genes examined (Figure 1A).

We next assessed whether the expression of the aforementioned genes correlates with the survival of patients with prostate cancer. To this end, we analyzed the TCGA dataset in the Prostate Integrative Expression Database (Figure 1B). Patient survival was independent of *MCL1* gene expression. Similarly, the gene expression of *HUWE1*, *BTRC*, or *FBXW7* did not affect patient survival. However, patients with high *USP9X* expression displayed significantly shorter survival than patients with low *USP9X* expression.

Although MCL1 expression did not affect the survival of patients with prostate cancer, expression data suggests a more prominent role for the deubiquitinase USP9x. We, therefore, hypothesized that the deubiquitinase USP9x contributes to prostate cancer progression by controlling Mcl-1 protein stability. To address this question, we analyzed Mcl-1 and USP9x protein levels in human prostate samples classified by pathologists into benign prostatic tissue (Mcl-1: n = 7, USP9x: n = 18), low-grade prostate cancer (Gleason score 6–7, Mcl-1: n = 12, USP9x: n = 18), high-grade prostate cancer (Gleason score 8–10, Mcl-1: n = 14, USP9x: n = 17), and progressive prostate cancer of patients receiving palliative care to relieve pain (Mcl-1: n = 18, USP9x: n = 23). Patients’ characteristics are summarized in Table 1. In benign tissue, Mcl-1 was hardly detectable, while USP9x immunoreactivity was detected in epithelial structures only (Figure 2A). Mcl-1 and USP9x immunoreactivity increased with prostate cancer progression (Figure 2A–C). Of these examined tissue probes, 45 prostate cancer samples and 10 benign prostate tissue samples were used to correlate Mcl-1 and USP9x levels (Figure 2D). Especially at advanced prostate cancer stages, increasing numbers of specimens displayed increased USP9x immunoreactivity together with increased Mcl-1 immunoreactivity. The moderate and highly significant positive correlation between Mcl-1 and USP9x immunoreactivity (Spearman’s r = 0.560, *p* = 0.000043; Pearson’s r = 0.510, *p* = 0.00025) proposed an interrelation between the anti-apoptotic protein Mcl-1 and the deubiquitylating enzyme USP9x in prostate cancer.

### 3.2. Mcl-1 Protein Levels Change in LNCaP and PC3 Prostate Cancer Cells after Irradiation

To examine the interaction between the anti-apoptotic protein Mcl-1 and USP9x in more detail, we employed LNCaP and PC3 prostate cancer cell lines. In particular, we addressed whether this interaction in prostate cancer cells could affect the response to therapies, particularly to radiotherapy.

To examine radiation-induced cytotoxicity in LNCaP and PC3 cells, apoptosis levels were determined by quantification of DNA fragmentation (subG1 cell fraction) using flow cytometry following irradiation with 10 Gy (Figure 3A and Appendix A). Increased cell populations of LNCaP and PC3 cells with fragmented DNA were detected 24 h after irradiation. The apoptosis rate continued to increase until 72 h after irradiation. 72 h after irradiation, around 10% of LNCaP and 15% of PC3 cells displayed DNA fragmentation. Apoptosis induction was verified by analyzing the cleavage of caspase-3 and caspase-3 substrate poly (ADP-ribose)-polymerase PARP, both of which are indicative of caspase activation (Figure 3B). We detected weak caspase-3 and PARP cleavage in PC3 cells 48 h and 72 h after irradiation. In contrast, we did not detect any caspase-3 and PARP cleavage in irradiated LNCaP cells.

Ionizing radiation might also facilitate non-apoptotic cell death without caspase activation [39]. Therefore, we quantified cell death after irradiation after analyzing the dissipation of mitochondrial membrane potential (ΔΨm low), which is indicative for all dying cells irrespective of the cell death mode (Figure 3C and Appendix A). In the non-irradiated control group, more dead cells were detected by determining ΔΨm dissipation than DNA fragmentation. Irradiation with 10 Gy increased cell death by 10–15% after 72 h.

Our experiments demonstrate that ionizing radiation modestly induced cell death in both prostate cancer cell lines, with apoptosis contributing to radiation-induced cell death in PC3 cells but hardly in LNCaP cells.

### 3.3. Downregulation of Mcl-1 Protein Levels Sensitizes LNCaP and PC3 Prostate Cancer Cells to Ionizing Radiation-Induced Cell Death

Bcl-2 family members play an important role in the maintenance of mitochondrial homeostasis and cell death induction, especially during the activation of the intrinsic apoptosis pathway [40,41]. We, therefore, analyzed the protein levels of several Bcl-2 family members, including Mcl-1, in both prostate cancer cell lines (Figure 3D). Levels of pro-apoptotic proteins Bak, Bim, and Noxa were similar in both cell lines. LNCaP cells expressed higher levels of pro-apoptotic Bax and of both anti-apoptotic proteins Mcl-1 and Bcl-2 than PC3 cells. In contrast, PC3 cells displayed higher levels of anti-apoptotic Bcl-xL than LNCaP cells. We next examined whether protein levels of the different Bcl-2 family members changed after irradiation with 10 Gy (Figure 3E). Levels of anti-apoptotic protein Bcl-xL, as well as pro-apoptotic proteins Bax, Bak, Puma, and Bim, did not change in both cell lines following irradiation. However, levels of anti-apoptotic Bcl-2 declined while levels of anti-apoptotic Mcl-1 increased in LNCaP cells after irradiation. In irradiated PC3 cells, in contrast, Mcl-1 levels decreased, while levels of Noxa, an interacting partner of Mcl-1, increased. Our data demonstrate that ionizing radiation regulates levels of specific Bcl-2 family members in a cell-specific manner.

The radiation-induced change in the anti-apoptotic proteins might influence the threshold level for apoptosis induction and cell survival. Particularly the change in Mcl-1 protein levels can alter sensitivity to radiotherapy. Therefore, we evaluated cell death following Mcl-1 mRNA silencing in both LNCaP and PC3 prostate cancer cells, using specific siRNA pools at a concentration between 10 nM and 100 nM (Figure 4A). 48 h after transfection with siRNA, we detected downregulation of Mcl-1 protein, which becomes more effective with increasing siRNA concentrations. The most effective knockdown in LNCaP cells to 24% was observed when 100 nM Mcl-1 siRNA was used. In PC3 cells, 10 nM Mcl-1 siRNA was enough to reduce the Mcl-1 protein level to 19%. DNA fragmentation to evaluate apoptosis and dissipation of ΔΨm to examine cell death were quantified 48 h after transfecting cells with respective Mcl1 siRNA concentrations or non-targeting siRNA (Figure 4B,C and Appendix A). Apoptosis and cell death in LNCaP cells increased with rising concentrations of Mcl1 siRNA. Transfection with 100 nM Mcl-1 siRNA induced cell death in 45% of LNCaP cells after 48 h. Although Mcl-1 knockdown was more effective in PC3, apoptosis and cell death were induced modestly and only after transfection with the highest Mcl-1 siRNA concentrations. We, therefore, conclude that cell survival depends on high Mcl-1 levels, particularly for LNCaP cells.

For the next set of experiments, we transfected prostate cancer cells with 50 nM Mcl-1 or non-targeting (nt) control siRNA 24 h prior to irradiation with 10 Gy. 48 h after irradiation, DNA fragmentation and ΔΨm dissipation 48 h were quantified (Figure 4D,E). After Mcl-1 knockdown, we detected a significant increase in radiation-induced apoptosis and cell death in both LNCaP and PC3 prostate cancer cells. Thus, the downregulation of Mcl-1 sensitized prostate cancer cells to ionizing radiation.

### 3.4. Ionizing Radiation Affects Protein Stability of Mcl-1 in Prostate Cancer Cells

Unequal knockdown efficiencies in the two prostate cancer cell lines might result from varying Mcl-1 stabilities. Changes in the Mcl-1 degradation rate could even explain the accumulation of Mcl-1 in LNCaP cells and the downregulation of Mcl-1 in PC3 cells after irradiation. We, therefore, analyzed Mcl-1 protein stability in LNCaP (Figure 5A) and PC3 cells (Figure 5B) in non-irradiated cells and 48 h after irradiation with 10 Gy. To this end, the decline of Mcl-1 protein was analyzed following treatment with 2 µM cycloheximide (CHX), an inhibitor of protein translation. The calculated half-life time of Mcl-1 increased significantly from 70 min in non-irradiated LNCaP cells to 101 min 48 h after irradiation (Figure 5A). In contrast, Mcl-1 protein levels were lower in PC3 cells 48 h after irradiation (Figure 5B), and the calculated half-life time of Mcl-1 decreased from 52 min to 39 min 48 h after irradiation, although the decrease was not significant (*p* = 0.06). A direct comparison of Mcl-1 protein stability in non-irradiated cells revealed a significantly shorter Mcl-1 half-life time in PC3 than in LNCaP cells (Figure 5C).

Previous studies linked the phosphorylation of Mcl-1 at serine 139 and threonine 163 to enhanced ubiquitylation and degradation, while the deubiquitylating enzyme USP9x is able to remove polyubiquitin chains from Mcl-1, thereby preventing Mcl-1 degradation at proteasome and stabilizing Mcl-1 protein levels [22,34]. Although the Mcl-1 level in PC3 cells was lower than in LNCaP cells, the USP9x level in PC3 cells was similar to that in LNCaP cells (Figure 5D). We next examined the phosphorylation of Mcl-1 at serine 159/threonine 163 and USP9x levels after irradiation with 10 Gy (Figure 5E). In LNCaP cells, we detected phosphorylated Mcl-1 as a band around 40 kDa, the predicted molecular weight. This signal decreased with time after irradiation, coinciding with increased Mcl-1 level. In contrast, in PC3 cells, we detected a feint phospho-Mcl-1 band at 40 kDa, which increased 48 h after irradiation. In addition, we detected high molecular weight phospho-Mcl-1 bands of more than 100 kDa, which accumulated after irradiation with 10 Gy, coinciding with decreased Mcl-1 level at 40 kDa. Interestingly, USP9x levels in LNCaP and PC3 cells did not change after irradiation.

Taken together, Mcl-1 was less stable in PC3 cells than in LNCaP cells due to increased phosphorylation and degradation. Moreover, in LNCaP cells, Mcl-1 became dephosphorylated following irradiation resulting in less degradation and stabilization of Mcl-1, while increased phosphorylation of anti-apoptotic Mcl-1 in PC3 cells after irradiation resulted in its enhanced degradation.

### 3.5. Knockdown of USP9x Expression Improves Response to Radiotherapy

Although USP9x protein levels remained unchanged following irradiation (Figure 5E), the enzyme activity or the interaction with Mcl-1 could have changed after irradiation. To clarify the role of USP9x in the regulation of Mcl-1 stability and cell survival, we knocked down USP9x expression using 50 nM USP9x siRNA. An efficient USP9x knockdown in both cell lines was confirmed 48 h after transfection in both cell lines (Figure 6A). 48 h after transfection, we detected slightly increased cell fractions with fragmented DNA (Figure 6B) and dissipated ΔΨm (Figure 6C), suggesting elevated cell death levels in response to USP9x knockdown due to apoptosis. Cell death and apoptosis levels did not alter much at 72 h and 96 h after transfection with USP9x siRNA. Next, we irradiated prostate cancer cells with 0 or 10 Gy 48 h after transfection with USP9x siRNA. DNA fragmentation (Figure 6D) and dissipation of ΔΨm (Figure 6E) were examined 48 h after irradiation (96 h after transfection). Similar to Mcl-1 knockdown, the knockdown of USP9x resulted in increased radiation-induced DNA fragmentation and dissipation of ΔΨm in both cell lines. Moreover, examination of Mcl-1 levels 48 h after irradiation with 0 Gy and 10 Gy revealed that USP9x knockdown prevented the radiation-induced increase of Mcl-1 levels in LNCaP cells, while only minor effects of USP9x knockdown on MCL-1 levels were detected in irradiated PC3 cells (Figure 6F). Analyzing the clonogenic survival, we detected a lowered survival fraction (SF) after irradiation when USP9x expression was silenced using RNAi technology (Figure 6G). As a consequence of USP9x knockdown, the radiosensitizing effect was more pronounced in LNCaP than in PC3 cells.

In summary, our experiments revealed that deubiquitinase USP9x is able to regulate sensitivity to radiotherapy by increasing Mcl-1 levels, thereby elevating the threshold for apoptosis induction in prostate cancer cells.

### 3.6. Bcl-2 and Bcl-xL Regulate Prostate Cancer Cell Survival in Cooperation with Mcl-1

So far, our results demonstrated that USP9x controls Mcl-1 protein levels and, in this way, regulates the response to radiotherapy in prostate cancer cells. High Mcl-1 protein levels, however, seem to be more important for LNCaP than for PC3 cell survival. Herein, the anti-apoptotic proteins Bcl-2 and Bcl-xL might influence the threshold level for apoptosis induction and cell survival independently of Mcl-1 [42]. To address this question, we transfected LNCaP and PC3 cells with 50 nM cells with 50 nM Mcl-1 or non-targeting control siRNA 24 h prior to treatment with 1 µM ABT-263, a small molecule specifically inhibiting Bcl-2 and Bcl-xL but not Mcl-1. 48 h after treatment with ABT-263, apoptosis (Figure 7A) and cell death levels (Figure 7B) were determined as described before. Treatment with 1 µM ABT-263 alone hardly induced apoptosis, whereas downregulation of Mcl-1 induced apoptosis only in LNCaP but not in PC3 cells. Treatment with 1 µM ABT-263 after Mcl-1 knockdown killed almost all cells. These results demonstrate that treatment with ABT-263 at non-toxic concentrations was able to push prostate cancer cells into cell death when Mcl-1 was additionally neutralized.

Next, we inhibited the anti-apoptotic proteins Bcl-2 and Bcl-xL by ABT-263 using concentrations between 0.4 µM and 10 µM to examine at which dosis ABT-263 was able to induce cell death. 48 h after treatment with ABT-263, apoptosis (Figure 7C, black bars) and cell death levels (Figure 7D, black bars) were determined. In both LNCaP and PC3 cells, a significant increase in apoptosis and cell death was detected only after treatment with 4–10 µM ABT-263 (black bars). Treatment of LNCaP cells with 10 µM ABT-263 resulted in about 70% apoptotic and a similar number of dead cells, while treatment of PC3 cells with 10 µM ABT-263 facilitated apoptosis in around 25% and cell death in around 50% of cells. Finally, we irradiated LNCaP and PC3 cells with 10 Gy directly before treatment with ABT-263 at concentrations between 0.4 µM and 10 µM (Figure 7C,D, white bars). When co-irradiated, treatment with 0.4 µM, ABT263 was sufficient to increase apoptosis and cell death significantly. However, apoptosis and cell death levels were always lower in PC3 than in LNCaP cells. We concluded that PC3 cells reacted more refractory to ABT-263-induced apoptosis than LNCaP cells. Furthermore, our data demonstrate that inhibition of Bcl-2 and Bcl-xL by ABT-263 lowered the cell death threshold in prostate cancer cells, thus rendering both LNCaP and PC3 prostate cancer cells more susceptible to apoptosis induced by ionizing radiation.

## 4. Discussion

Anti-apoptotic Bcl-2 family members are critical regulators of intrinsic apoptosis as well as sensitivity to radiotherapy [32,33,39,41]. Among these protective proteins, Mcl-1 is particularly tightly regulated on transcriptional, translational, and posttranslational levels [20,21,22]. Here, we evaluated the change of Mcl-1 protein levels in human tissue samples during prostate cancer progression as well as the underlying mechanism responsible for the change of Mcl-1 protein levels in two prostate cancer cell lines. We provide evidence that transcriptional upregulation of the *MCL1* gene is not the cause of elevated Mcl-1 protein levels in prostate cancer. We detected increased protein levels of deubiquitinase USP9x, an enzyme known to increase Mcl-1 protein stability, during prostate cancer progression, which was positively correlated with Mcl-1 levels. A possible interaction between Mcl-1 and USP9x was examined in LNCaP and PC3 prostate cancer cells. Higher Mcl-1 protein stability was associated with higher Mcl-1 protein levels in LNCaP cells in comparison to PC3 cells. In addition, we presented convincing data that radiotherapy might affect Mcl-1 stability and, consequently, cellular Mcl-1 protein levels. Finally, we demonstrated that a knockdown of USP9x reduced Mcl-1 levels and sensitized particularly LNCaP cells to radiotherapy.

Our findings suggest that deubiquitinase USP9x is responsible for upregulated Mcl-1 protein levels, thereby contributing to prostate cancer progression and therapy resistance.

### 4.1. Regulation of MCL1 Gene Expression and Mcl-1 Protein Stability in Prostate Cancer

Deregulated apoptosis pathway is a hallmark of cancer [43]. Mechanisms preventing efficient apoptosis induction are also major causes of the poor response of cancer cells to radiotherapy. Acquired treatment resistance by prostate carcinoma cells was associated with apoptosis evasion. In this regard, members of the Bcl-2 protein family controlling apoptosis at the mitochondrial level are fundamental factors regulating the sensitivity of cancer cells to radiotherapy and chemotherapy [9,33,44]. Anti-apoptotic Bcl-2 family members are overexpressed in many tumor entities, including prostate cancer, and have been linked to detrimental survival [11,12,45]. Although all these proteins protect from apoptosis in a similar way, their regulation occurs in individual ways and might differ between tissues [46]. Particularly the regulation of Mcl-1 can be distinguished from that of Bcl-2 or Bcl-xL. Mcl-1 is characterized by a rapid turnover, resulting in usually low protein levels in normal tissue. However, the rapid degradation is often deregulated, leading to high Mcl-1 protein levels in tumor tissues [22,28,46]. In accordance with these observations, we detected no change in *MCL1* gene expression between normal prostate tissue and prostate cancer tissue, but we observed upregulated Mcl-1 protein levels during prostate cancer progression. Particularly advanced prostate cancer tissues very often display high Mcl-1 protein levels. While *USP9X* gene expression was slightly but insignificantly lower in prostate cancer tissue compared to normal prostate tissue, USP9x protein levels also increased during prostate cancer progression. Although the analysis of gene expression did not allow a distinction between different stages during prostate cancer progression, the gene expression data do not match with our results obtained from the immunohistological detection of USP9x. However, it is possible that the regulation of mRNA and protein levels changes during prostate cancer progression. Our immunohistological analysis suggests that translational or posttranslational regulation of Mcl-1 protein levels becomes more important during prostate cancer progression. Since high Mcl-1 protein levels often coincided with high USP9x protein levels at higher stages during prostate cancer progression, a posttranslational stabilization by USP9x provides here a credible explanation for elevated Mcl-1 levels. USP9x-mediated stabilization of Mcl-1 protein levels by preventing its degradation was described before as a mechanism that increased apoptosis threshold in other tumor entities [22,33]. We confirmed this stabilizing effect of USP9x on Mcl-1, particularly in LNCaP prostate cancer cells.

Furthermore, we detected that high *USP9X* gene expression correlated with a worse outcome in patients with prostate cancer, although *USP9X* gene expression was slightly but insignificantly lower in prostate cancer tissues than in normal tissue. High *USP9X* gene expression was associated before with shorter survival of patients with follicular lymphoma, esophageal squamous cell carcinoma, non-small cell lung cancer, and osteosarcoma [22,47,48,49], but not with the survival of patients with prostate cancer. Interestingly, we detected an association between patient survival and gene expression of *USP9X* only, but not of HUWE1, BTRC, and FBXW7, even though all gene products are able to interact with Mcl-1, further emphasizing the role of the deubiquitinase in prostate cancer progression.

Previous publications demonstrated an interaction of the deubiquitinase USP9x with different proteins, thereby not only modifying their turnover rate but also affecting their subcellular localization or activity [50,51,52,53]. During prostate cancer progression, the function of USP9x—as well as its specificity and affinity to its interacting partner molecules—might change, thereby affecting the survival of patients with prostate cancer. Although not all prostate cancer cells might use USP9x to stabilize Mcl-1, USP9x might account for elevated Mcl-1 levels, particularly in advanced prostate cancer. On the other hand, many prostate samples display high USP9x and low Mcl-1 immunoreactivities, suggesting an alternative role for USP9x besides the stabilization of Mcl-1 protein levels. In this context, previous publications demonstrated that USP9x-dependent stabilization of insulin receptor substrate 2 or the transcription factors PBX homeobox 1 and ERG supported prostate cancer growth and metastasis [51,52,53].

### 4.2. Increased Mcl-1 Stability by USP9x Mediates Prostate Cancer Resistance to Radiotherapy

One of the first options in the treatment of low-grade prostate cancer still residing in the prostate gland and in the treatment of high-grade prostate cancer that has grown into nearby tissues is ionizing radiation applied as curative therapy [2]. At early stages, prostate cancer cells respond well to radiotherapy and induce cell death in response to irradiation. At advanced stages, prostate cancer reacts more refractory to radiotherapy due to upregulated protective mechanisms allowing cancer cells to elope cell death induction after irradiation. Even when prostate cancer is categorized as incurable, ionizing radiation is employed as palliative therapy to keep cancer in check and to prevent or relieve symptoms caused by advanced prostate cancer [54,55]. In the treatment of prostate cancer, radiation therapy is usually applied in fractionated doses at 1.8 to 2.5 Gy per fraction to a total of 50 to 80 Gy [56]. In our experiments with LNCaP and PC3 cells, we used single doses of 1–5 Gy for long-term experiments and a single irradiation dose of 10 Gy to analyze effects up to 72 h after irradiation. Even at such a high dose as 10 Gy, radiation-induced cell death did not exceed 15%.

To improve the response to radiotherapy particularly in advanced prostate cancer, it is important to identify the mechanism that protects prostate cancer cells from radiation-induced cell death and, in a second step, to target these protective mechanisms. We evaluated the role of Mcl-1 and USP9x in response to irradiation with 10 Gy employing LNCaP and PC3 cell lines. PC3 cells, previously described as androgen-receptor-negative, express lower androgen receptor levels than LNCaP cells and are a model for castration-resistant androgen-independent prostate cancer cells, while LNCaP cells serve as a model for androgen-dependent prostate cancer cells [57]. We provided evidence that regulation of Mcl-1 stability is a factor determining the response of prostate cancer cells to radiotherapy. We summarized the regulation of Mcl-1 protein levels in Figure 8. Mcl-1 stability and Mcl-1 protein levels were lower in PC3 than in LNCaP cells. Irradiation resulted in decreased Mcl-1 stability in PC3 cells, while Mcl-1 stability increased in LNCaP cells. Previous publications described glycogen synthase kinase (GSK)-3β-dependent phosphorylation of Mcl-1 serine 159/threonine 163, facilitating its ubiquitylation and degradation via proteasome [58,59]. We detected increased Mcl-1 phosphorylation in irradiated PC3 cells that went together with accumulation of phosphorylated Mcl-1 at a higher molecular range, suggesting a phosphorylation-driven polyubiquitylation of Mcl-1 that targets the anti-apoptotic protein for proteasomal degradation. The reduced Mcl-1 half-life time we detected in irradiated PC3 cells supports this conclusion. In LNCaP cells, in contrast, we detected less phospho-Mcl-1 and no accumulation at a higher molecular weight. In this context, it was demonstrated that tyrosine kinase inhibitors increased Mcl-1 degradation [42], highlighting the importance of phosphorylation for Mcl-1 stability. We concluded that specific factors counteract Mcl-1 phosphorylation, its polyubiquitylation and its subsequent degradation in LNCaP cells.

We previously identified USP9x as a factor regulating radiosensitivity in glioblastoma cells [33]. Downregulation of USP9x by siRNA in LNCaP cells prevented Mcl-1 stabilization, increased apoptosis induction and reduced clonogenic survival following irradiation, particularly in LNCaP but to a lesser extent also in PC3 cells. Thus, we now confirmed USP9x as a radioprotective factor in prostate cancer cells. The different extent of radioprotection in LNCaP and PC3 cells can be ascribed to additional factors. Among others, radiation-induced activation of USP9x might additionally facilitate the accumulation of Mcl-1 in LNCaP but not in PC3 cells. Moreover, decreased kinase activity and/or increased phosphatase activity could be responsible for less phosphorylated Mcl-1 in irradiated LNCaP cells and, in this way, might contribute to Mcl-1 stability [42]. Finally, additional factors controlling Mcl-1 stability might be differentially expressed and regulated in irradiated LNCaP and PC3 cells [29,30,60,61].

### 4.3. Bcl-2, Bcl-xL, and Mcl-1 Cooperatively Protect Prostate Cancer Cells from Apoptosis

Previous publications described Bcl-2 and Bcl-xL as protective factors controlling apoptosis independently of Mcl-1 [42,62]. We now verified the cooperative protection from apoptosis by Bcl-2, Bcl-xL, and Mcl-1 in LNCaP and PC3 prostate cancer cells. We previously demonstrated that neutralization of anti-apoptotic family members sensitized glioblastoma, lung, and colorectal cancer cells to radiation-induced cell death [33,41]. The radiosensitizing effect of the Bcl-2/Bcl-xL inhibitor ABT-263 depended on the neutralization or downregulation of Mcl-1. In prostate cancer cells, however, ionizing radiation resulted in decreased Mcl-1 protein levels in PC3 cells but increased Mcl-1 levels in LNCaP cells. After irradiation, decreased Mcl-1 levels might reduce the anti-apoptotic threshold and render PC3 cells more susceptible to ABT-263-induced apoptosis. In this context, previous publications demonstrated that the sensitivity to ABT-263-induced apoptosis was prevented by high Mcl-1 levels [60,63,64]. In addition, we detected increased levels of the BH3-only protein Noxa coinciding with decreasing Mcl-1 levels in irradiated PC3 cells. Upon binding to Mcl-1, Noxa can promote Mcl-1 degradation, thus lowering cellular Mcl-1 levels [65,66]. However, the increased Mcl-1 levels in irradiated LNCaP cells did not increase further resistance to ABT-263-induced apoptosis. In contrast, although ionizing radiation increased Mcl-1 levels in irradiated LNCaP cells, these cells reacted more sensitively to ABT-263-induced apoptosis. The increased sensitivity to ABT-263 in irradiated LNCaP cells might result from decreased Bcl-2 protein levels. LNCaP cells expressed higher levels of Bcl-2 than PC3 cells. Anti-apoptotic Bcl-2 might sequester pro-apoptotic Bcl-2 family members [10,67]. Thus, the downregulation of Bcl-2 levels might sensitize LNCaP cells to apoptosis by fostering the release of the pro-apoptotic proteins from sequestration and inducing MOMP.

In addition, ABT-263 improved cancer cell cytotoxicity in androgen-dependent prostate cancer cells undergoing androgen-deprivation therapy and interfered with progression to androgen-independent prostate cancer, demonstrating a potent effect of Bcl-2/Bcl-xL inhibition at early stages of prostate cancer [68,69]. However, ABT-263 was less effective in androgen-resistant prostate cancer cells when applied alone [69]. After treatment with ABT-263, androgen-dependent LNCaP cells induced cell death more efficiently than androgen-independent PC3 cells, supporting the previous observations.

Of note, survival of LNCaP cells strongly depends on Mcl-1 and Bcl-2/Bcl-xL, as neutralization of Mcl-1 by siRNA and additional inhibition of Bcl-2 and Bcl-xL by ABT-263 induced more efficient cell death in LNCaP than in PC3 cells. Since PC3 cells express even lower levels of Bax than LNCaP cells, an insufficient activation of Bax in PC3 cells might prevent effective MOMP and cytochrome C release into cytosol and subsequent caspase activation. However, the comparison of cell death and apoptosis levels induced in PC3 cells after Mcl-1 knockdown and additional Bcl-2/Bcl-xL inhibition suggest apoptosis inhibition downstream of MOMP. Such changes might include the phosphorylation of cytochrome C, resulting in decreased binding to apoptotic peptidase activating factor-1 (APAF-1) and subsequent formation of the apoptosome complex, or high levels of inhibitors of apoptosis that interfere with caspase activation [70,71,72,73].

### 4.4. Limits of the Study

Finally, we want to point out that our experiments analyzing USP9x-mediated Mcl-1 stability were carried out in two prostate cancer cell lines only. Comparing both cell lines, we demonstrated that Mcl-1 protein stability correlated with Mcl-1 protein levels before and after irradiation. However, we did not examine mRNA levels and translational efficiency. Both can contribute to the changes of Mcl-1 protein levels and may vary in the extent of contribution between different cell lines as well as the stages during prostate cancer progression. Moreover, we did not examine additional factors regulating Mcl-1 stability in either cell lines or tissue samples during prostate cancer progression. Even when Mcl-1 might protect from cell death, particularly apoptosis, many other factors might affect cell death induction downstream of Mcl-1 and MOMP, thus playing a more important role than Mcl-1 in the regulation of apoptosis and radiosensitivity in other prostate cancer cell lines or during prostate cancer progression. Although our results suggest that the posttranslational control of Mcl-1 stability becomes more important during prostate cancer progression, more experiments in prostate cancer tissues or primary prostate cancer cells are still required to substantiate the role of Mcl-1 and USP9x as factors responsible for a poor response to radiotherapy during prostate cancer progression.

## 5. Conclusions

We demonstrated a concurrent upregulation of Mcl-1 and USP9x protein levels during prostate cancer progression. Furthermore, we provided insights into the mechanism by which USP9x regulates prostate cancer cell survival. Our data revealed that USP9x increased the protein stability of anti-apoptotic Mcl-1, resulting in the accumulation of the protective protein, particularly in LNCaP cells. Moreover, we provided evidence that radiotherapy might affect Mcl-1 stability. Targeting Mcl-1 or USP9x improved the response of LNCaP and PC3 cells to radiotherapy and might prove beneficial for prostate cancer patients receiving radiotherapy.

## Figures and Tables

**Figure 1 cancers-15-02496-f001:**
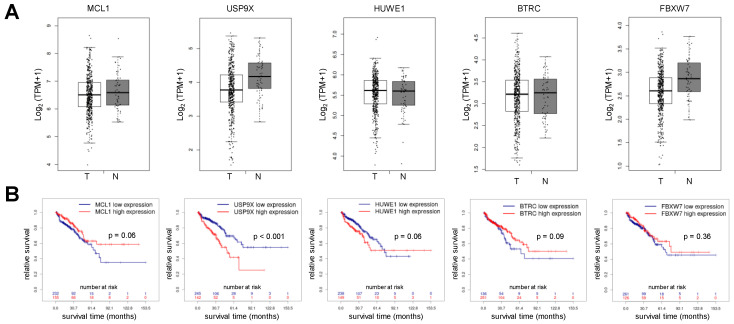
Gene expression of *MCL1* is not altered in prostate cancer and does not correlate with patient survival. (**A**) Differential gene expression analysis of *MCL1*, *USP9X*, *HUWE1*, *BTRC* and *FBXW7* demonstrates no significant gene expression (*p* > 0.01) between prostate adenocarcinoma tissue (T: 492 samples) and normal tissue (N: 52 samples). Analysis was performed using GEPIA database (http://gepia.cancer-pku.cn, accessed on 7 March 2023). Box plots show median gene expression (horizontal line in the box), interquartile range above quartile 1 to quartile 3 (box). The whiskers show gene expression within the lowest quartile (<Q1) and highest quartile (>Q3). (**B**) Gene expression of *MCL1*, *HUWE1*, *BTRC* and *FBXW7* in prostate cancer tissue does not correlate with patient survival, but high gene expression of *USP9X* in prostate cancer tissue correlated with shorter overall survival. The relationship between gene expression (*MCL1* or *USP9X)* and survival was assessed using TCGA dataset at Prostate Integrative Expression Database (https://pixdb.org.uk/PIXdb/pages/index.php, accessed on 7 March 2023). Survival curves of high-expressing and low-expressing cohorts and number of patients at risk at respective time points are presented in Kaplan-Meier plots.

**Figure 2 cancers-15-02496-f002:**
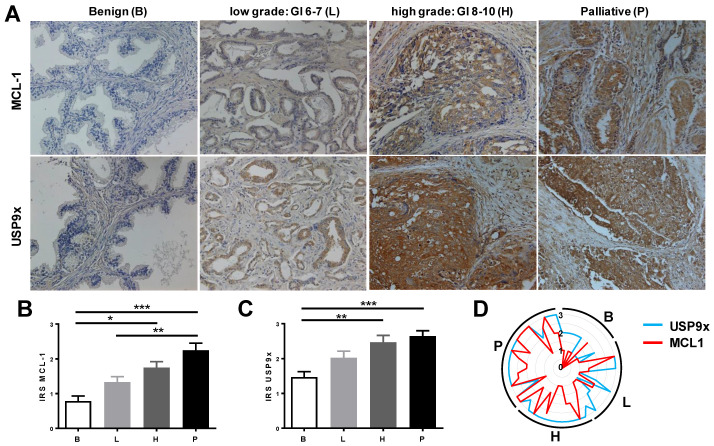
Mcl-1 and USP9x protein levels increase during prostate cancer progression. Prostate cancer tissues were analyzed by immunohistochemistry using antibodies against Mcl-1 and USP9x. The following number of tissues were analyzed: benign prostate tissue (B: Mcl-1: n = 7, USP9x: n = 18), low-grade prostate cancer (L: Gleason score 6–7, Mcl-1: n = 12, USP9x: n = 18), high-grade prostate cancer (H: Gleason score 8–10, Mcl-1: n = 14, USP9x: n = 17), and advanced prostate cancer of patients receiving palliative care to relief pain (P: Mcl-1: n = 18, USP9x: n = 23). (**A**) Representative tissue staining of matching samples employing anti-Mcl-1 and anti-USP9x antibodies. Staining intensity (immunoreactivity score IRS) of MCL-1 (**B**) and USP9x (**C**) increased during prostate cancer progression. Data shows mean values ± S.E.M., *: *p* < 0.05; **: *p* < 0.01; ***: *p* < 0.001. (**D**) 56 samples (B: 10 samples; L: 14 samples; H: 12 samples; P: 20 samples) showing co-expression of Mcl-1 and USP9x.

**Figure 3 cancers-15-02496-f003:**
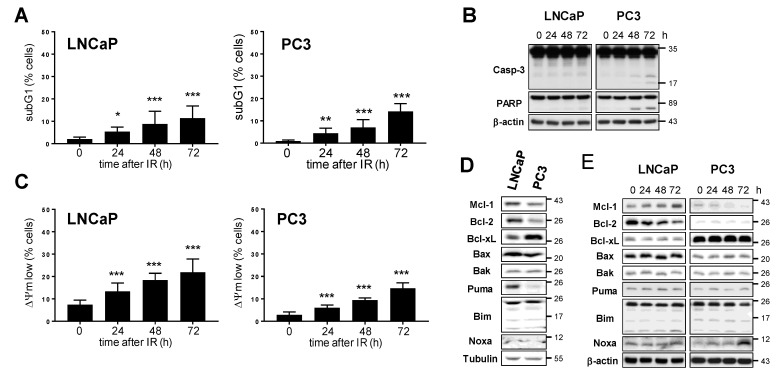
Prostate cancer cells induce moderate cell death after exposure to ionizing radiation. LNCaP and PC3 cells were irradiated with 10 Gy. At indicated time points after irradiation, DNA fragmentation (**A**) and dissipation of mitochondrial membrane potential ΔΨm (**B**) were analyzed by flow cytometry. (**C**) At indicated time points after irradiation, cells were lysed. Processing of caspase-3 and its substrate PARP were analyzed by Western blot. β-actin was used as loading control. (**D**) Protein levels of different Bcl-2 family members were compared in LNCaP and PC3 cells. Tubulin was used as loading control. (**E**) To examine radiation-induced changes of Bcl-2 family members, cells were lysed at indicated time points after irradiation and analyzed by Western blot. β-actin was used as loading control. Data shows mean values ± S.D., n = 3, *: *p* < 0.05; **: *p* < 0.01; ***: *p* < 0.001. Significance was calculated to respective non-irradiated control cells at t = 0 h. The original western blot images can be found in File S1.

**Figure 4 cancers-15-02496-f004:**
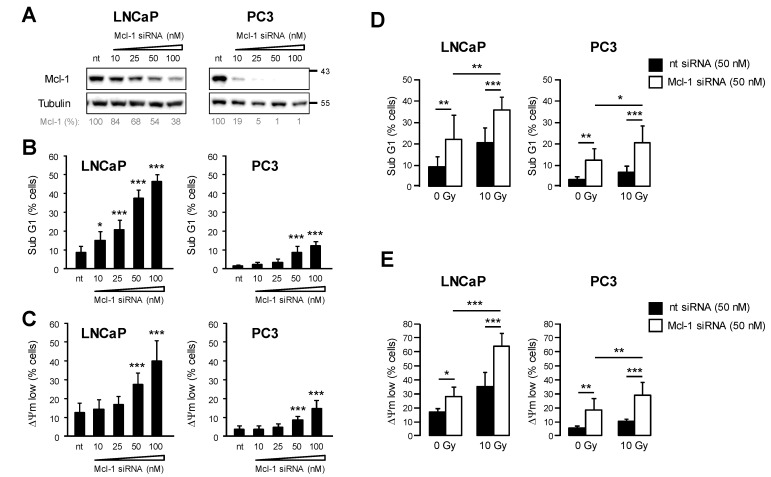
Knockdown of Mcl-1 sensitized prostate cancer cells to radiation-induced apoptosis. (**A**–**C**) LNCaP and PC3 cells were transfected with different concentrations of Mcl-1 siRNA ranging from 10 nM to 100 nM (final concentration in the medium). 100 nM non-targeting siRNA (nt) was used as control. (**A**) 48 h after transfection, downregulation of Mcl-1 protein level was verified by Western blot. 48 h after transfection, DNA fragmentation (**B**) and dissipation of mitochondrial membrane potential ΔΨm (**C**) were analyzed by flow cytometry. Data shows mean values ± S.D., n = 3, *: *p* < 0.05; ***: *p* < 0.001. Significance was calculated to respective cells transfected with 100 nM nt siRNA. (**D**,**E**) LNCaP and PC3 cells were transfected with 50 µM of Mcl-1 or nt siRNA. 24 h later, cells were irradiated with 0 Gy or 10 Gy. 48 h after irradiation, DNA fragmentation (**D**) and dissipation of mitochondrial membrane potential ΔΨm (**E**) were analyzed by flow cytometry. Data shows mean values ± S.D., n = 3, *: *p* < 0.05; **: *p* < 0.01; ***: *p* < 0.001. The original western blot images can be found in File S1.

**Figure 5 cancers-15-02496-f005:**
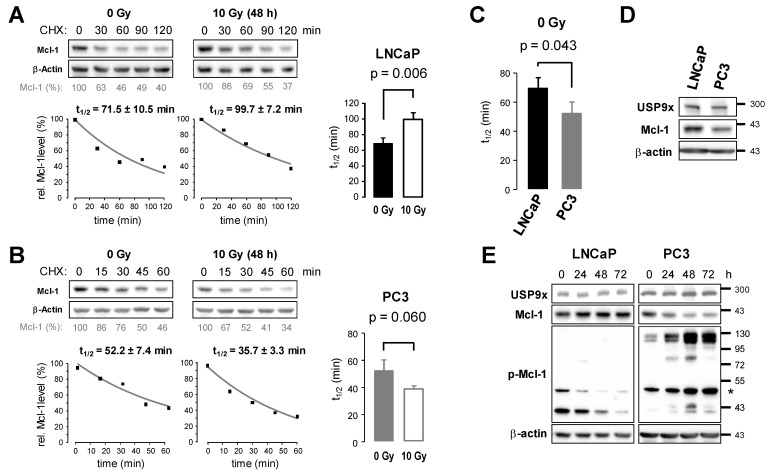
Mcl-1 stability increased in LNCaP cells but decreased in PC3 cells after irradiation. LNCaP (**A**) and PC3 (**B**) cells were irradiated with 0 Gy or 10 Gy. 48 h after irradiation, cells were treated with 2 µM cycloheximide (CHX) for indicated time. Then, cells were lysed, and the samples were analyzed by Western blot using an antibody against Mcl-1 and β-actin. After densitometric analysis, Mcl-1 levels were normalized to respective β-actin levels before normalization to respective controls (0 min CHX). A representative densitometric analysis is shown below the respective Western blots. The normalized levels were used to calculate Mcl-1 half-life time t_1/2_ (right panel). Data shows mean values ± S.D. (**C**) Comparison of Mcl-1 half-life time in non-irradiated LNCaP and PC3 cells. (**D**) Lysates of non-irradiated LNCaP and PC3 cells were analyzed by Western blot to compare MCL-1 and USP9x protein levels. Β-actin was used as loading control. (**E**) To detect radiation-induced changes of Mcl-1, phospho-Mcl-1 (Ser159/Thr163) and USP9x, LNCaP and PC3 cells were lysed at indicated time points after irradiation and lysates were examined by Western blot. β-actin was used as loading control. The original western blot images can be found in File S1.

**Figure 6 cancers-15-02496-f006:**
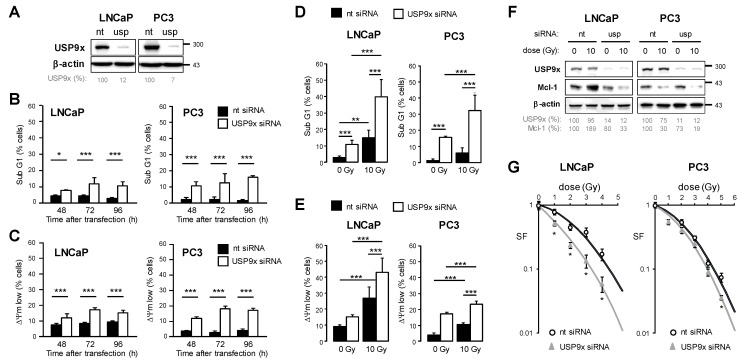
Knockdown of deubiquitylating enzyme USP9x sensitized increased radiation-induced cytotoxicity in prostate cancer cells. LNCaP and PC3 cells were transfected with 50 nM USP9x siRNA or non-targeting (nt) siRNA. (**A**) 48 h after transfection, cells were lysed, and USP9x levels were examined by Western blot. β-actin was used as loading control. At indicated time points after transfection, DNA fragmentation (**B**) and dissipation of mitochondrial membrane potential ΔΨm (**C**) were analyzed by flow cytometry. Data shows mean values ± S.D., n = 3, *: *p* < 0.05; ***: *p* < 0.001. (**D**–**F**) 48 h after transfection, cells were irradiated with 0 Gy or 10 Gy. 48 h after irradiation, DNA fragmentation (**D**) and ΔΨm (**E**) were analyzed by flow cytometry. Data shows mean values ± S.D., n = 3, **: *p* < 0.01; ***: *p* < 0.001. (**F**) 48 h after irradiation, cells were lysed, and changes in Mcl-1 and USP9x protein levels were examined by Western blot. β-actin was used as loading control. Densitometric analysis is shown below the blots. (**G**) 48 h after transfection, cells were irradiated with 0–5 Gy and cells were incubated to allow growth of colonies from single cells. After counting colonies for each condition, surviving fraction (SF) was calculated. Data shows mean values ± S.D., n = 3, *: *p* < 0.05. The original western blot images can be found in File S1.

**Figure 7 cancers-15-02496-f007:**
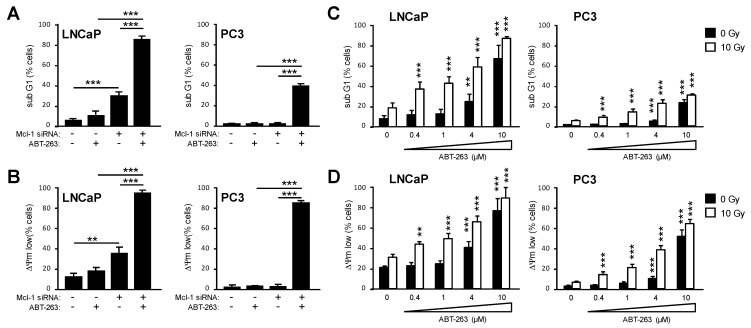
Inhibition of Bcl-2 and Bcl-xL by ABT-263 induces apoptosis in cooperation with MCL-1 knockdown and sensitizes prostate cancer cells to radiation-induced apoptosis. (**A**,**B**) LNCaP and PC3 cells were transfected with 50 nM Mcl-1 siRNA. 24 h later, cells were treated with 1 µM ABT-263 for 48 h. DNA fragmentation (sub G1, (**A**)) and dissipation of ΔΨm (ΔΨm low, (**B**)) were examined by flow cytometry. (**C**,**D**) LNCaP and PC3 cells were irradiated with 0 Gy or 10 Gy. Immediately after irradiation, cells were treated with ABT-263 at concentrations ranging from 0.4 µM to 10 µM. 48 h after irradiation, DNA fragmentation (sub G1, (**C**)) and dissipation of ΔΨm (ΔΨm low, (**D**)) were analyzed by flow cytometry. Data shows mean values ± S.D., n = 3, **: *p* < 0.01; ***: *p* < 0.001. Significance was calculated to respective solvent-treated cells (0 µM ABT-263) until otherwise indicated.

**Figure 8 cancers-15-02496-f008:**
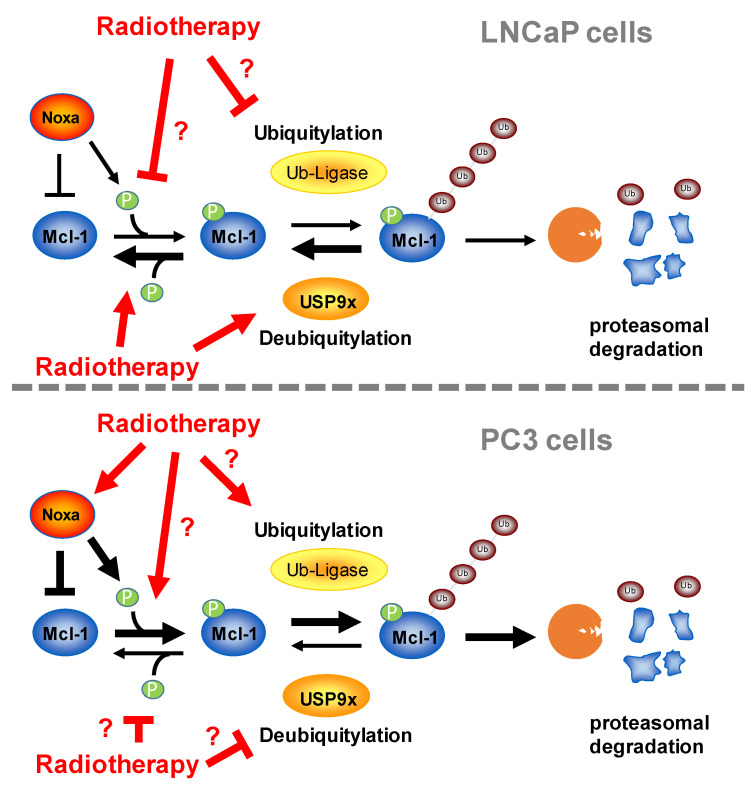
Regulation of Mcl-1 protein levels in LNCaP and PC3 prostate cancer cells. Mcl-1 protein levels are higher in LNCaP cells than in PC3 cells due to reduced polyubiquitylation and reduced proteasomal degradation. Upper panel: After irradiation, less Mcl-1 is phosphorylated and polyubiquitylated, resulting in increased Mcl-1 protein stability in LNCaP cells. Deubiquitinase contributes to increased Mcl-1 protein stability, but it remains unclear whether irradiation also blocks ubiquitylation of Mcl-1. It also remains unclear whether inhibition of phosphorylation or increased dephosphorylation is responsible for decreased phospho-Mcl-1 levels in LNCaP cells after irradiation. Lower panel: Mcl-1 protein stability is decreased in PC3 cells after irradiation due to increased polyubiquitylation and proteasomal degradation. Radiotherapy results in upregulated Noxa levels, thus facilitating Noxa-dependent polyubiquitylation and degradation of Mcl-1. It remains unclear whether radiotherapy also increased ubiquitin ligase activity and protein levels or blocked deubiquitylating activity. In the regulation of Mcl-1 levels, USP9x plays a less important role in PC3 cells than in LNCaP cells. Radiotherapy also increased phospho-Mcl-1 levels, but it is unclear to what extent radiotherapy facilitates phosphorylation and/or decreased dephosphorylation.

**Table 1 cancers-15-02496-t001:** Baseline patient demographics and histopathology (n = 74). 32 patients underwent radical prostatectomy (n = 15 for high Gleason score (GL) and n = 17 for low Gleason score specimen), palliative transurethral resection of the prostate (n = 26 for palliative specimen) and radical cystoprostatectomy, transvesical prostatectomy or transurethral resection (n = 16 for benign specimen).

Characteristics	Sub-Characteristics	Value (n %)
Age (range)		68 ± 7.3
Surgical procedures	radical prostatectomy	32 (44%)
	palliative transurethral resection	26 (35%)
	radical cystoprostatectomy	6 (8%)
	transvesical prostatectomy	4 (5%)
	transurethral resection	6 (8%)
Stage (pT) (n = 32 for Gleason scored specimen)	pT1	0
	pT2	23 (72%)
	pT3	9 (28%)
Lymph node metastasis (pN) (n = 58; without benige specimen)	pN0	30 (52%)
	pN1	13 (22%)
	pNX	15 (26%)
Distant metastasis (M) (n = 58; without benige specimen)	M0 (high and low GL)	32 (57%)
	M1 (all palliative)	20 (33%)
	MX (palliative)	6 (10%)
USP9x scores (epithelium)	benige	16 (21%)
	low	16 (21%)
	high	16 (21%)
	palliative	25 (33.6%)
Mcl-1 scores (epithelium)	benige	7 (9%)
	low	12 (16%)
	high	14 (19%)
	palliative	22 (29%)

## Data Availability

Data can be obtained upon request from J.R. (justine.rudner@uk-essen.de) or S.H.M. (stephan.huber@uni-tuebingen.de).

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
