# Peer review of "Protein Levels of Anti-Apoptotic Mcl-1 and the Deubiquitinase USP9x Are Cooperatively Upregulated during Prostate Cancer Progression and Limit Response of Prostate Cancer Cells to Radiotherapy"

_cancers, 2023, doi:10.3390/cancers15092496_

Round 1

Reviewer 1 Report (Previous Reviewer 2)

Resubmitted manuscript bears deficiencies similar to the first submission.

1.     Title overstates the results.  Title states: “Posttranslational Control of Mcl-1 Protein Levels by Deubiquitinase USP9x Constitutes a Critical Mechanism Regulating Cancer Cell Survival during Prostate Cancer Progression” however, cell survival during prostate cancer progression was not assessed.

2.    Simple summary blurs distinction between results obtained from experiments in cell lines and from the analysis of clinical samples.  Simple summary states: “Mcl-1 and USP9x protein levels were cooperatively upregulated during prostate cancer progression but were not associated with gene expression. More-23 over, radiotherapy itself is able to increase Mcl-1 protein stability. Targeting USP9x reduced Mcl-1 protein levels and thus improved the response to radiotherapy.”  This statement can be interpreted as the result of samples analysis from patients who underwent radiation therapy, however, in reality only cell lines were subjected to gamma-irradiation.

3.    Abstract overstates novelty of the role of USP9X in regulating MCL-1 levels in prostate cancer.  Abstract states: “Moreover, we identified the deubiquitinase USP9X as a factor regulating Mcl-1 levels in prostate cancer cells, thus limiting cytotoxic response to radiotherapy. 

The role of USP9X in regulating MCL-1 levels in prostate cancer has been demonstrated earlier.

See refs

20.  Trivigno, D.; Essmann, F.; Huber, S.M.; Rudner, J. Deubiquitinase USP9x confers radioresistance through stabilization of Mcl-709 1. Neoplasia (New York, N.Y.) 2012, 14, 893-904.

48.  Zhang C, Cai TY, Zhu H, Yang LQ, Jiang H, Dong XW, Hu YZ, Lin NM, He QJ, Yang B. Synergistic antitumor activity of gemcitabine and ABT-737 in vitro and in vivo through disrupting the interaction of USP9X and Mcl-1. Mol Cancer Ther. 2011 Jul;10(7):1264-75. doi: 10.1158/1535-7163.MCT-10-1091. Epub 2011 May 12. PMID: 21566062

4.    There is abundant literature on the role of MCL-1 in prostate cancer cells apoptosis that should be cited in the introduction. 

5.    Results section states correlation between MCL- 1 and USP9x in prostate tumor samples.  This analysis should be supported by original data showing levels of MCL-1 and USP9x in the same tissue sample.

6.    Alternative method for quantifying MCL- 1 and USP9x proteins in prostate tumor samples should be used to support IHC data.

Author Response

Reviewer 2 Report (New Reviewer)

This study is about the Radiosensitivity of prostate cancer. The author have collected some PCa sample to determine the relationship of Mcl-1 and USP9x. The cellular experiments also indicated that the expression of Mcl-1 and USP9x are related to radiosensitivity of prostate cancer. However, there are some questions mentioned by the previous reviewers have not been clarified clearly.

According to the reviewer 1, how USP9x regulated Mcl-1 is not display in the manuscript. Whether the USP9x interacted with Mcl-1 should be determined by co-immunoprecipitation assay. In addition, whether the expression of Mcl-1 would be downregulated by USP9x also didn’t shown in manuscript.

There is another recommendation provided by reviewer 1 is that the author should show a graphic abstract to make this study more understandable.  

The title of this manuscript mentioned that the posttranslational of Mcl-1 regulated by Deubiquitinase USP9x, however, which type of posttranslational of Mcl-1 was regulated by USP9x have not been mentioned in manuscript.

The representative image of  FASC should be presented in the figure (for example n figure 3A and 3C and figure 4B and 4C.

Author Response

Reviewer 3 Report (New Reviewer)

In the manuscript “Posttranslational Control of Mcl-1 Protein Levels by Deubiquitinase USP9x…”  Hogh-Binder and colleagues focused their investigations on the protein levels of anti-apoptotic Mcl-1 and its stabilizer deubiquitinase, USP9x. The research includes human tissue specimens, representative of stages of prostate cancer. Importantly, the authors assessed the posttranslational regulation of Mcl-1 by USP9x at protein levels, their relevance to cancer promoting mechanisms and potential modulation of this cancer promoting mechanism in LNCaP and PC3 prostate cancer cell lines.

Through a meta-analysis the authors show that USP9x gene mRNA expression levels correlate with shorter disease progression and provided compelling evidence that none of the other examined gene expressions (potentially affecting the pathway: MCL1, HUWE1, BTRC and FBXW7) shows disease progression correlations. Thus, the research further is focused on the protein level/turnover and cancer biology of McL1-USP9x interaction. The authors demonstrate that Mcl-1 and USP9x protein levels are increased in human prostate specimens with higher disease stages. These findings imply increases in radiation resistance along the lines of disease stages. The authors point out correctly that obtaining human prostate cancer specimens in a post-radiation setting is not a common practice in standard of care. Thus, the authors are focusing on mechanistic models of prostate cancer radiation resistance and cancer biology.

A baseline induction of cell death was measured in response to radiation in AR positive LNCaP and AR negative PC3 cells, noting that McL1 knockdown is able to sensitizes LNCaP and PC3 cell lines to radiation-induced apoptosis. Along these lines the authors demonstrate that radiation can stabilize the protein levels of anti-apoptotic Mcl-1. However, the authors note that in the AR negative (X-chromosome deleted) PC3 cell line Mcl-1 was less stable when compared to the AR positive (mutant AR) LNCaP cells owning differences in phosphorylation and protein turnover issues. This observation is consistent with the basal/luminal characteristics of these cell line, respectively.

A key fining of this manuscript is that knockdown of USP9x sensitizes the radiation induced cell death by increasing Mcl-1 protein levels. Placing in the context of established radiation resistance mechanisms the authors show cooperation between McL-1 and Bcl-2/ Bcl-xL in survival of the examined prostate cancer cells.

Overall, this is a logically structured research article that adds new knowledge to the mechanisms of escape mechanisms in response to radiation therapy of prostate cancer.

Some updates of the cited article are suggested, in particular the review article citations where newer reviews are now available.

Author Response

Reviewer 4 Report (New Reviewer)

Paper entitled “Posttranslational Control of Mcl-1 Protein Levels by Deubiquitinate USP9x Constitutes a Critical Mechanism Regulating Cancer Cell Survival during Prostate Cancer Progression” is a good one. The study design should be acknowledged because the authors based their research hypothesis on the analysis of available data about gene expression in patients. Especially they show the significance of USP9x by Kaplan-Meyer graphs. The authors present coherent results described in a clear way. In general this paper should be accepted – although it require some issues to be resolved by authors.

1.     Authors compared the level of expression of MCL1 gene in normal and cancerous samples. The difference in the number of those groups is very high. Authors should address correctly this issue. The question remains – although with the statistical test used by authors, there is no statistically significant difference between normal and cancerous patients for USP9x, some tendency is observed which could be of course biased by variation is sample size. In any case please describe what are box whiskers etc.? Information about tests and exact p values should be provided.

2.     Is dose 10 Gy the most optimal according to real one used during therapy? Please justify it and address it to the proper literature.

3.     Authors perform numerous experiments and as mentioned present results in a coherent way indicating the mechanism of USP9x-dependent regulation of apoptosis in prostate cancer cell lines. It will beneficial to add figure summarizing results. At scheme please indicate both mechanisms which are based on your research as well as factors that still need to be investigated (i.e. different expression of unrecognized factors controlling expression of  Mcl1 between LNCaP and PC-3 cell lines – 591 line.)

4.     While authors have tested only two prostate cancer cell lines they should be more critical about the overall significance of their results in the context of prostate cancer. Especially those lines are cell lines delivered from lymph nodes and bone metastasis of prostate cancer. Please try to underline it more in discussion - it does not neglect work done by authors in any case but make the presentation of results more fair and clear.

Author Response

Reviewer 5 Report (New Reviewer)

Corrections are appropiate.

Round 2

Reviewer 1 Report (Previous Reviewer 2)

Resubmitted manuscript provides more balanced interpretation of the results.

It is recommended that authors state whether finding of correlation between USP9x gene expression and survival in prostate cancer patients is novel, and whether this association was assessed in other cancers.

There are some statements that need to be “tamed down” to reflect presented results.  For example: “Our findings suggest that increased expression of deubiquitinase USP9x upregulates MCL-1 protein in prostate cancer cells and may contribute to prostate cancer progression and therapy-resistance” would be more appropriate than “Our findings identified anti-apoptotic protein Mcl-1 and the deubiquitinase USP9x AS DECISIVE FACTORS IN PROSTATE CANCER PROGRESSION” (page 14, line 498). 

Author Response

This manuscript is a resubmission of an earlier submission. The following is a list of the peer review reports and author responses from that submission.

Round 1

Reviewer 1 Report

The authors investigated the relevance of anti-apoptotic Bcl-2, Bcl-xL, and Mcl-1 for radiation-induced cytotoxicity in human LNCaP and PC3 prostate cancer cells. This study sounds interesting, but it is difficult to understand the relationship between the analysis of Bcl-2, Bcl-xL, and ABT-263 and the analysis of Mcl-1 and USP9x. The reviewer recommends several points to improve this manuscript.

#1 What is the association between the Bcl-2/Bcl-xL/ABT-263 in-22 and  Mcl-1/USP9x? Too many players in one study make unclear what is the main purpose of this study. Please provide some schemas to understand the total story of this study.   

#2 The relationship between the ABT-263 and Mcl-1/USP9x is unclear. ABT-263 inhibits anti-apoptotic Mcl-1/USP9x?

#3 The authors need to evaluate radiation-resistant prostate cancer tissues to confirm the present findings. Without those data, the conclusions of this study are just cellular experimental results and are not highly credible. Figure 6 shows that Mcl-1 and USP9x levels increase during prostate cancer progression. but it is difficult to support the hypothesis (radiation-resistant phenotype) of this study.

Reviewer 2 Report

Two prostate cancer cell lines LNCaP and PC3 were investigated for their sensitivity to ionizing radiation and expression of BCL2-family proteins.

BCLXL antagonist ABT263 and knockdown of MCL-1 by shRNA increased sensitivity of both cell lines to radiation-induced apoptosis.

Similar effect was observed after knockdown of USP9x, a deubiquitinase that was previously shown to deubiquitinate MCL-1.  As expected, knockdown of USP9x decreased MCL-1 expression. 

Analysis of prostate tissue by immunohisotchemistry (IHC) was used to assess the expression levels of MCL-1 and USP9x levels.  Based on this analysis authors report increased expression of both proteins along with progression of prostate cancer.

The presented experimental data do not support the conclusion stated in the manuscript title (Upregulation of Deubiquitylating Enzyme USP9x During Prostate Cancer Progression is Responsible for Acquisition of Mcl-1-Mediated Radioresistance) as radioresistance of prostate tumors was never assessed.  There is no information provided whether patients from whom prostatectomy samples were collected received radiotherapy.  Reliability of (IHC) data is also questionable, as results were not confirmed by an independent method.